# Functional Recovery and Regenerative Effects of Direct Transcutaneous Electrical Nerve Stimulation in Treatment of Post-COVID-19 Guillain–Barré and Acute Transverse Myelitis Overlap Syndrome: A Clinical Case

**DOI:** 10.3390/jfmk9010040

**Published:** 2024-02-26

**Authors:** Mustafa Al-Zamil, Natalia G. Kulikova, Inessa A. Minenko, Numman Mansur, Denis M. Zalozhnev, Marat B. Uzdenov, Alina A. Dzhanibekova, Alikhan A. Gochiyayev, Natalia A. Shnayder

**Affiliations:** 1Department of Physiotherapy, Faculty of Continuing Medical Education, Peoples’ Friendship University of Russia, 117198 Moscow, Russia; kulikovang777@mail.ru (N.G.K.); d-64-158@mail.ru (N.M.); 2Department of Restorative Medicine and Neurorehabilitation, Medical Dental Institute, 127253 Moscow, Russia; kuz-inna@mail.ru (I.A.M.); zalozhnev@mail.ru (D.M.Z.); 3Department of Sports Medicine and Medical Rehabilitation, I.M. Sechenov First Moscow State Medical University, 119991 Moscow, Russia; 4City Clinical Hospital Named after V. V. Vinogradov, 117292 Moscow, Russia; 5Medical Institute, North Caucasian State Academy, 369001 Cherkessk, Russia; uzdenov1@rambler.ru (M.B.U.); mardjan2011@yandex.ru (A.A.D.); hanali2002@mail.ru (A.A.G.); 6Institute of Personalized Psychiatry and Neurology, V.M. Bekhterev National Medical Research Centre for Psychiatry and Neurology, 192019 Saint Petersburg, Russia; 7Shared Core Facilities “Molecular and Cell Technologies”, Professor V. F. Voino-Yasenetsky Krasnoyarsk State Medical University, 660022 Krasnoyarsk, Russia

**Keywords:** acute transverse myelitis, autoimmune demyelination, COVID-19, Guillain–Barré syndrome, transcutaneous electrical nerve stimulation, TENS, spinal cord injuries

## Abstract

Transcutaneous electrical nerve stimulation (TENS) has proven effective in treating pain in many experimental and clinical studies. In addition to the analgesic effect, direct TENS of peripheral nerves had anti-inflammatory and regenerative effects in the treatment of distal polyneuropathy and spinal cord injury. This work demonstrates the experience of using direct TENS in the treatment of a 52-year-old patient with post-COVID-19 Guillain–Barré (GBS) and acute transverse myelitis (ATM) overlap syndrome. Direct TENS of peripheral nerves showed high efficiency in enhancing the therapeutic effect of combined plasma exchange and pharmacotherapy by 89.5% with a significant reduction in neuropathic pain, motor and sensory deficits, bladder and bowel disorders and regression of neurophysiological changes. We suggest that direct TENS of peripheral nerves can be a promising option for combined therapy of GBS and ATM overlap syndrome and other diseases with the simultaneous development of distal polyneuropathy and spinal cord injury. Further trial studies are required.

## 1. Introduction

Acute transverse myelitis (ATM) is one of the rare spinal cord injuries that most often induces bilateral segmental demyelination or necrosis of one part of the spinal cord at one or more levels [1,2]. The complete pathophysiology of ATM is unknown, but many authors believe that the disease develops as a result of an autoimmune reaction, which may be idiopathic or associated with post-infectious or para-infectious events [3]. Recently, several clinical cases of ATM associated with post-COVID-19 infection have been reported [4,5,6] and, in extremely rare cases, in combination with another autoimmune disease—Guillain–Barré syndrome (GBS) [7,8]. In 40% of patients with ATM and in 30% of patients with GBS, residual motor, sensory and autonomic disorders may persist for months or even years due to resistance to pharmacotherapy [9,10]. In this regard, in order to enhance the therapeutic effect in the treatment of these diseases, it is recommended to use combined methods of treatment in the initial stages [11]. In clinical practice, the most common physiotherapy treatments for ATM and GBS were spinal cord stimulation [12,13], transcranial magnetic stimulation [14], electroacupuncture [15] and transcutaneous electrical nerve stimulation (TENS) [16].

TENS was originally intended for the treatment of pain [17]. Gradually, the effectiveness of TENS in accelerating regenerative processes and reducing neurological deficits has been proven in many clinical and experimental studies [18,19,20,21,22].

TENS is a therapy for diseases of peripheral nerves and is implemented through many mechanisms. The anti-inflammatory effect of TENS is due to the cytokine reduction pathway and reduction of proinflammatory cytokines in the blood [23] and in tissues [24]. The sensory recovery after TENS treatment is associated with improvements in somatosensory evoked potentials, TNF-α and synaptophysin expression in the dorsal root ganglion, somatosensory cortex and hippocampus [25]. The motor recovery effect of TENS has been studied in numerous experimental and clinical studies. Many authors believe that the use of TENS leads not only to accelerated functional motor recovery but also to an enhancement in the axon quantity and the diameter of the regenerated axons [18]. At the same time, low-frequency high-amplitude TENS (LF TENS), rather than high-frequency low-amplitude TENS (HF TENS), leads to an increase in fiber diameter and thickening of the myelin sheath [26].

In addition to the treatment of peripheral nervous system diseases, direct TENS has found application in the management of patients with brain and spinal cord disorders. It is suggested that the pathogenetic basis of the central antispastic effect of TENS is a decrease in neuronal excitability, a decrease in the sensitivity of the stretch reflex [27,28] and an increase in presynaptic inhibition [29]. Clinically, in patients with central paralysis, direct TENS was reported to be effective in the regression of motor deficit of ankle dorsiflexion after direct stimulation of the common peroneal nerve [30] and in the recovery of motor function of the hand after direct stimulation of the median nerve [31]. The functional motor recovery effect of direct TENS may be due to the modulation of the excitability of the corticospinal tracts [32], a decrease in short-term intracortical inhibition and an increase in the corticomuscular connection [33].

Several studies have demonstrated the effectiveness of direct TENS in the treatment of GBS and spinal cord injuries. However, simultaneous treatment of ATM and GBS with direct peripheral nerve TENS has never been performed, and this opportunity was used to demonstrate the antinociceptive, anti-inflammatory, functional recovery and regenerative effects of using direct TENS in the treatment of a patient with post-COVID-19 GBS and ATM overlap syndrome.

## 2. Case Presentation

### 2.1. Patient History

A 52-year-old Caucasian man arrived to our attention with prominent paraparesis of the lower extremities, sensory disorders in the legs and hands and bladder and bowel dysfunction. In the anamnesis, the first symptom of the disease was listed as a burning sensation in the right leg, which appeared acutely 14 days after the coronavirus infection (COVID-19). Subsequently, the neurological deficit progressed rapidly over 7 days. Post-COVID-19 GBS and ATM overlap syndrome was diagnosed based on clinical, laboratory, electroneuromyography and MRI findings. The patient was hospitalized 4 days after onset of neurological symptoms. Inpatient treatment included intravenous immunoglobulin therapy, plasma exchange and standard pharmacotherapy. There was a significant regression of sensory, motor and autonomic disorders. However, for 17 days after discharge, neuropathic pain, paresthesia, hypoesthesia in the upper and lower extremities, as well as lower paraparesis, and minor bladder and bowel disorders persisted. In this regard, he was referred to continue outpatient physiotherapy treatment at the Brain and Spine Clinic of the Moscow region (Table 1).

### 2.2. Before Direct TENS Treatment

#### 2.2.1. Physical Examination

(1)Higher nervous activity: Normal waking consciousness was established. No cognitive impairments were registered, but a focus on reducing self-criticism was found.(2)Motor functions: In the lower extremities, paraparesis was noted in proximal and distal muscles with an average of 3/5 points on the right side and 4/5 points on the left. More pronounced paraparesis was recorded in ankle dorsiflexion with a decrease of up to 2/5 points on the right side and up to 3/5 points on the left. No motor deficit was noted in the hands and face. No muscle atrophy was detected. A mixed type of gait disorder was disclosed (ataxic, spastic and paretic). Carporadial, flexion elbow and extensor elbow reflexes were symmetrically reduced. Patellar reflexes were hyperactive, more so on the right side, but Achilles reflexes diminished, especially on the left side. Babinski reflex and variations of the Babinski reflex, such as the Schaeffer, Oppenheim, Gordon and Chaddock reflexes, were evoked on the right side, and only the Babinski reflex was evoked on the left side.(3)Muscle tone was slightly decreased in the upper extremities, symmetrically, but asymmetrically increased in the lower extremities, with greater spasticity in the right leg. The severity of muscle tone according to the Modified Ashworth Scale was 2 in the left leg and 3 in the right leg. Nocturnal painful spasticity was assessed on a 10-point scale: 7 points in the right leg and 5 points in the left leg.(4)Sensory functions: No sensation abnormality was found on the face, torso or arms. Temperature, touch and pinprick perception were reduced in the upper extremities in a glove-type fashion. The level of temperature hypoesthesia in the upper extremities was detected in the middle third of the forearm and below (4/5), and tactile hypoesthesia was detected in the lower third of the forearm and below (3/5) in both hands. All sensations, including vibration and position perception, were reduced below the L2 lumbar level and were more pronounced on the right half of the body. The decrease in sensation was assessed by the patient on a 10-point scale compared to sensation on the face and was 10 points on the right side and 7 points on the left side.(5)Pain syndrome on the Visual Analog Scale (VAS) was 3 points in the hands and 6 points in the legs. Paresthesia (symptoms of numbness, burning and tingling) was assessed on a 10-point scale: 5 points in the hands and 7 points in the legs.(6)Bladder and bowel dysfunction: Control over voluntary urination was not complete. The patient had difficulty urinating in small portions. Intermittent catheterization was performed every 8 h. Neurogenic bladder dysfunction was assessed by the specific health-related Russian-translated Short Form of Qualiveen (SF-Qualiveen) questionnaire and averaged 2.6/4.0 [34]. Despite the use of polyethylene glycol-based suppositories, bowel care time increased to 1–1:20 h. However, the patient did not require digital stimulation for bowel care sessions, but a mini-enema was administered on average once a week. The frequency of bowel movements was 1–3 times a week. No fecal incontinence was observed. The severity of bowel dysfunction assessed by the Neurogenic Bowel Dysfunction Score was 8/47 [35].

#### 2.2.2. Laboratory Results

The results of a complete blood count were within normal limits. No abnormalities were detected in routine biochemical and hematological studies, including serum levels of 25OH(D3), cyanocobalamin, thyroid hormones (T3, T4, TSH), glycated hemoglobin and creatine phosphokinase, and serological tests for hepatitis B and C, human immunodeficiency virus and syphilis. Detection of neuromyelitis optica IgG was negative with titers less than 1/10 (normal range < 1:10). Twelve kinds of ganglioside autoantibody were measured, including GM1, GM2, GM3, GM4, GQ1b, GT1b, GT1a, GD1a, GD1b, GD2, GD3 and sulfatide. All these tests were negative. Serum immunoglobulin (IgG and IgM) of AGAs was tested. All these tests were negative. Anti-MOG, anti-aquaporin-4 (anti-AQP4, anti-cardiolipin (CL)/β2 and glycoprotein I (β2GPI) testing was negative. Tumor markers were within the normal range.

#### 2.2.3. Neuroimaging

(1)Brain MRI: There are no abnormal focal areas of altered signal intensity in the cerebral hemispheres, brainstem or cerebellum.(2)Cervical MRI: T1- and T2-weighted MRI images of the cervical spine showed signs of spondylosis, spondyloarthrosis and multiple extrusions of intervertebral discs at cervical level C5-D1 without compression of the dural sac and spinal cord. No spinal cord pathology was detected.(3)Lumbar MRI: T1- and T2-weighted MRI scans of the thoracic spine showed signs of spondylosis, spondyloarthrosis and multiple protrusion of intervertebral discs at lumbar level Th12-S1 without compression of the dural sac and spinal cord. No spinal cord pathology was detected.(4)Thoracic MRI: T2-weighted MRI images of the thoracic spine showed asymmetrical mild patchy high-signal changes in the spinal cord between levels D9 and D10, affecting the posterolateral sides of the spinal cord bilaterally, but predominantly on the right side with minimal swelling (Figure 1).

#### 2.2.4. Electrophysiology

##### Evoked Electroneurography

(1)In the upper extremities, electroneurography of median and ulnar nerves showed a slight decrease in motor conduction velocity, an increase in terminal latency of the compound muscle action potential (CMAP) and normal CMAP amplitudes and duration. Additional temporal dispersion in the elbow–wrist nerve segment was registered in an examination of ulnar nerves. A pronounced decrease in sensory nerve conduction velocity and sensory nerve action potential (SNAP) amplitude was registered in the median and ulnar nerves bilaterally. No decremental motor responses to repetitive nerve stimulation of the right axillary and median nerves were registered.(2)In the lower extremities, a significant decrease in motor conduction velocity of the common peroneal and tibial nerves was recorded with a marked decrease in CMAP amplitude and an increase in terminal latency and duration of CMAP bilaterally. Temporal dispersion was recorded in the ankle–knee segment of the tibial nerves and in the ankle–fibular head segment of the common peroneal nerves. SNAP of the bilateral superficial peroneal nerves, medial plantar nerves and sural nerves could not be recorded. No decremental motor responses to repetitive nerve stimulation of the right tibial nerve were registered.

##### Quantitative Needle Electromyography

Needle electromyography showed signs of denervation and reinnervation activity with evidence of ongoing denervation including fibrillation potentials and positive sharp waves in the tibialis anterior and gastrocnemius muscles bilaterally.

Evoked electroneurography and needle electromyography findings are more characteristic of acute inflammatory demyelinating polyneuropathy (AIDP) with secondary axonopathy.

##### Brain Evoked Potentials Tests

Visual and brainstem auditory response tests were within the normal limits.

### 2.3. Direct TENS Treatment

The patient underwent physiotherapy treatment using direct TENS. The treatment included 2 options of TENS: HF TENS and LF TENS.

#### 2.3.1. Equipment

The BTL-4000 smart/premium device was used for direct TENS (BTL Industries Ltd., Hertfordshire, UK) (registration number: FSZ 2010/06686, dated 29 April 2010, Russia).

#### 2.3.2. Characteristics of Current

(1)HF TENS: monopolar square-wave pulse was used in the stimulation with a 100 Hz frequency and 50 μs duration. The amplitude of stimulation was increased until a painless sensation of vibration was achieved.(2)LF TENS: monopolar square-wave pulse was used in the stimulation with a 1 Hz frequency and 200 μs duration. The amplitude of stimulation was increased until painless muscle contraction was achieved.

#### 2.3.3. Electrode Fixation

In LF TENS, the cathode was fixed over the proximal edge of the nerve, but the anode (pen form) was not fixed and moved to the area of the nerve projection distally (Figure 2). In HF TENS, the electrodes are replaced by a cathode instead of an anode (Table 1).

Direct TENS was carried out in two parallel courses. In the first course, the peroneal and tibial nerves were stimulated bilaterally. In the second course, stimulation of median, ulnar and radial nerves was performed on each side. The number of sessions in each course was 15. Each of these courses was conducted alternately every other day. The duration of each procedure did not exceed 40 min.

### 2.4. After Direct TENS Treatment

#### 2.4.1. Physical Examination

(1)Motor functions: Motor deficit completely regressed in the left leg with the exception of ankle dorsiflexion (strength = 4.5/5). In the right leg, slight residual motor deficits were registered proximally and distally (strength = 4/5). However, ankle dorsiflexion strength improved to a 3/5. A marked improvement in walking was noted with the preservation of elements of spasticity and steppage gait. Gait velocity and functional mobility increased in good lighting, but in the dark, some difficulties were noted. Asymmetrical dysmetria was detected during the heel-to-shin test, mainly in the right leg, but not during the finger-to-nose test. Compared to the previous examination before treatment, no significant changes in reflexes were detected. The Babinski reflex was evoked only on the right, with a total reduction in the Schaeffer, Oppenheim, Gordon and Chaddock reflexes and the absence of spontaneous activity. The muscle tone of the upper extremities was normalized. Moderate spasticity was detected in the right leg but not in the left. The severity of muscle tone according to the Modified Ashworth Scale was 0 in the left leg and 1 in the right leg. Nocturnal painful spasticity regressed to 3 points in the right leg and 1 point in the left.(2)Sensory functions: Temperature and tactile hypoesthesia in the upper extremities decreased significantly and was observed only on the fingers. Comparing the current status with the previous one, the patient began to distinguish tactile and painful stimuli on the right half of the body below the lumbar level of L2. No improvement was found in vibration and position perception. According to the patient’s assessment on a 10-point scale, hypoesthesia was 5 points in the right half and 3 points in the left. Pain syndrome in the hands completely decreased, but in the legs, it remained at 3 points. Paresthesia averaged 2 points in the hands and 3 points in the legs.(3)Neurogenic bladder dysfunction improved significantly and scored 1.25 on the SF-Qualiveen scale. With the use of polyethylene glycol-based suppositories, bowel care time decreased to 30–40 min. The severity of bowel dysfunction assessed by the Neurogenic Bowel Dysfunction Score decreased to 4 points.

#### 2.4.2. Electrophysiology

##### Evoked Electromyography

Electromyography of the upper and lower extremities was re-examined 2 months after direct TENS treatment (Table 2 and Table 3).

(1)In the upper extremities, the CMAP of the median and ulnar nerves remained normal. A slight increase in conduction velocity and a moderate decrease in distal latency were registered in motor nerves. In sensory nerves, an increase in the SNAP amplitudes of the median nerves was registered, with an average increase of 20.1%, and the ulnar nerves increased by 84.9%. At the same time, changes in conduction velocity were not significant.(2)In the lower extremities, significant increases in the CMAP of the peroneal and tibial nerves were recorded, especially in the right peroneal (Figure 3) and the left tibial nerves (Figure 4). Motor conduction velocity did not change in peroneal nerves. Electroneurography of the tibial nerves showed an increase in conduction velocity of 18% in the left nerve and 21% in the right. Terminal latency decreased to 19% in the left peroneal nerve and to 24% in the right. No improvement was noted in SNAP amplitude and sensory conduction velocity of peroneal, tibial and sural nerves.

#### 2.4.3. Neuroimaging

Asymmetric bilateral mild focal high-signal changes in the spinal cord were assessed between levels D9 and D10 on a T2-weighted MRI of the thoracic spine. Compared to the previous MRI examination before direct TENS treatment, there were no pronounced morphological dynamics (Figure 5).

## 3. Discussion

A 52-year-old male with spastic proximal and distal paraparesis of the lower extremities, ataxic, spastic and paretic gait disorder, hypoesthesia in the upper and lower extremities and below the L2 lumbar level, neuropathic pain and bladder and bowel disorders was under our observation. Symptoms developed rapidly, 14 days after the onset of COVID-19 infection. Based on clinical, laboratory, MRI and EMG findings, diagnosis of GBS and ATM overlap syndrome was suspected and confirmed. In his treatment, we used direct TENS of the peroneal, tibial, median, ulnar and radial nerves. TENS therapy was started 61 days after the onset of the disease and 32 days after the completion of plasma exchange. As a result, there was a significant improvement in the motor function of the lower extremities in the proximal muscles by an average of 30.5% and in the distal muscles by 43.8%. (Figure 6).

It is not possible to conclude whether recovery of motor function was due to improvement in upper or lower motor neuron function. However, it is known that in many studies, the direct TENS method is used in the treatment of flaccid [37] and spastic paralysis [38]. Nonetheless, to study the effectiveness of TENS in the treatment of both GBS and ATM, the symptoms of each disease were studied separately.

Dynamics of GBS symptoms: Hypoesthesia in the upper extremities was reduced by 75% in both hands. EMG showed an improvement in the SNAP amplitude of the median and ulnar nerves with no change in the conduction velocity of the sensory and motor nerves. An increase in CMAP amplitude of the peroneal and tibial nerves was recorded with a slight increase in motor neuron conduction velocity. Neuropathic pain was reduced in the upper extremities completely and by 50% in the lower extremities. The reduction in paresthesia averaged 60% in the upper extremities and 57% in the lower extremities.

Dynamics of ATM symptoms: Hypoesthesia below the L2 lumbar level regressed by 50% on the right side and by 57% on the left side. In the lower extremities, the severity of spasticity was regressed according to the Modified Ashworth Scale by 66.7% in the right leg and by 100% in the left leg. Nocturnal pain spasticity regressed by 66.7%. Of the five pathologic reflexes on the right foot, only the Babinski pathologic reflex can be evoked. In the left foot, the pathologic Babinski reflex is no longer evoked (Figure 7). In addition, reductions in bladder and bowel disorders of 51.9% and 50.0% were observed, respectively.

Thus, we found a significant regression of the neurological symptoms of polyneuropathy associated with GBS and the neurological disorders of myelopathy associated with ATM. It is important to note that improvement in conduction velocity studies of peripheral nerves indicates that the recovery of sensory and motor functions is not only functional in nature but is also the result of regenerative changes in the sensory and motor fibers of the nerves. Increased CMAP is more characteristic of functional recovery and axonal regeneration [39]. Further, the increase in conduction velocity and the decrease in terminal latency are reliable signs of nerve remyelination [40].

To determine the therapeutic effectiveness of TENS against the background of the therapeutic effect achieved after the use of plasma exchange and pharmacotherapy, neurological disorders were studied weekly from the onset of the disease to the end of observation. For ease of comparison, the average values of motor deficit, hypoesthesia and neuropathic pain were calculated after conversion to a 10-point system. The mean value of bladder and bowel dysfunction was also assessed after converting the SF-Qualiveen scores and Neurogenic Bowel Dysfunction scores into a 10-point system (Figure 8). A decrease in neurological disorders was noted in the lower extremities, upper extremities and bladder and bowel function on the 30th day after the use of plasma exchange with pharmacotherapy—23.6%, 39.7% and 34.65, respectively, and after the use of direct TENS compared with the results obtained on the 30th day after plasma exchange and pharmacotherapy—58.5%, 85.7% and 41.2%, respectively. Thus, we can come to the conclusion that TENS enhanced the therapeutic effect of the previously conducted course of plasma exchange with pharmacotherapy by 89.5%.

Many studies report that the long-term therapeutic effect after plasma exchange can be up to a year. However, the most pronounced therapeutic effect may develop within 30 days after the end of treatment [41,42].

In a number of studies, TENS has proven to be a highly effective method of treatment and rehabilitation for patients with GBS in the early stages of the disease [43,44]. We suggest that the therapeutic effect of direct TENS in the treatment of diabetic distal polyneuropathy is not significantly different from the therapeutic effect in the treatment of GBS. This effect is achieved by accelerating functional recovery, nerve fiber remyelination [25] and axonal regeneration [24] and increasing the regenerative capacity of peripheral nerves [45].

There are no published studies on the treatment of ATM using direct TENS, and only sporadic publications have focused on the effective management of this disease indirectly [16]. However, in the literature, more attention is paid to the treatment of spinal cord injuries with direct TENS [46]. In some studies, direct TENS of the tibial nerve was used to achieve bladder neuromodulation via modulation of autonomic nervous system functions [47] and to prevent neurogenic detrusor overactivity in patients with acute spinal cord injury [48]. The effectiveness of direct TENS can be explained by the stimulation of sub-lesioned neural circuits in the area of spinal cord injury, which still receives supraspinal and infraspinal inputs, but the excitability of sub-lesioned circuits is insufficient to maintain the function of impulse transmission between overlying and underlying areas. Activation of these neural circuits through electrical stimulation increases their excitability and leads to the restoration of connections with neighboring neurons above and below the damaged area [49,50].

It is important to note that the application of direct TENS as a stand-alone treatment may not explain all the reported benefits. This effect most likely results from the use of TENS in combination with pharmacotherapy and the late effect of plasma exchange. Moreover, it cannot be excluded that the cause of functional and neurophysiological improvement may be spontaneous biological regenerative processes, which are sometimes observed in acquired inflammatory polyneuropathies. However, after a detailed chronology of the disease and the dynamics of neurological disorders after treatment with TENS, it became obvious that the positive effectiveness of treatment occurred faster and with greater amplitude after the use of TENS.

## 4. Conclusions

Direct TENS of peripheral nerves showed high efficiency in enhancing the therapeutic effect of combined plasma exchange and pharmacotherapy in the treatment of a 52-year-old patient with post-COVID-19 Guillain–Barré and acute transverse myelitis overlap syndrome by 89.5%. We suggest that direct TENS of peripheral nerves can be a promising option for combining therapy of GBS and ATM overlap syndrome and other diseases with the simultaneous development of distal polyneuropathy and spinal cord injury. Further trial studies are required.

## Figures and Tables

**Figure 1 jfmk-09-00040-f001:**
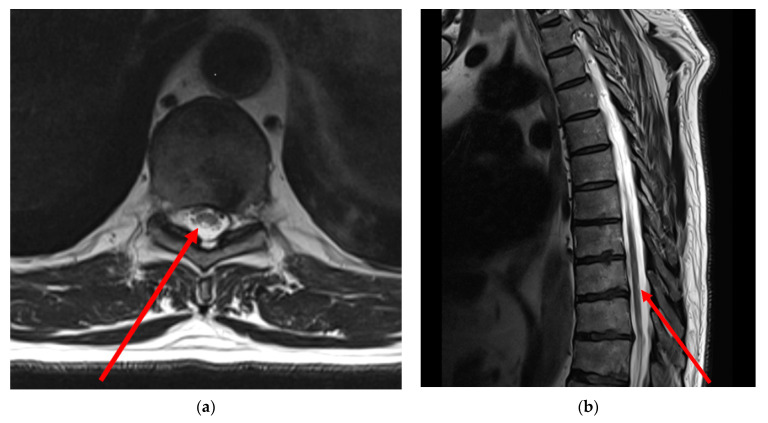
T2-weighted MRI images of the thoracic spine demonstrate incomplete segment of hyperintensity at thoracic Th9-Th10 level in axial scan (**a**) and in sagittal scan (**b**). Note: red arrows indicate the affected area of the spinal cord. Neuroimaging examination was performed 35 days after the onset of neurological symptoms.

**Figure 2 jfmk-09-00040-f002:**
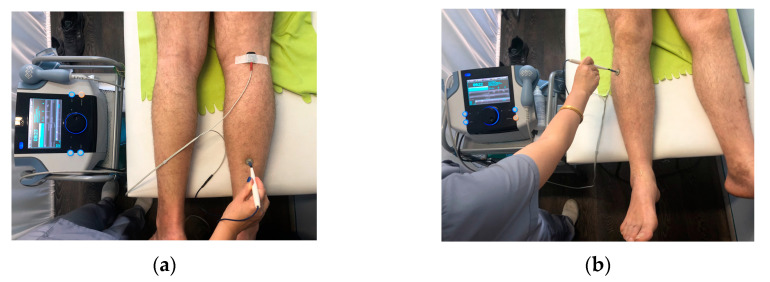
Technique of using low-frequency high-amplitude TENS of the right tibial (**a**) and right peroneal (**b**) nerves. Note: the red electrode is the cathode; the pen-form electrode is the anode.

**Figure 3 jfmk-09-00040-f003:**
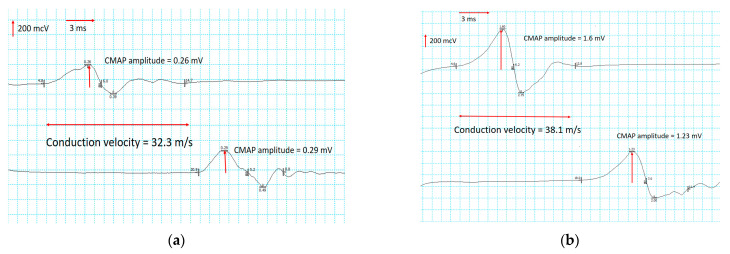
Nerve conduction study of the left tibial nerve before (**a**) and after (**b**) direct TENS treatment.

**Figure 4 jfmk-09-00040-f004:**
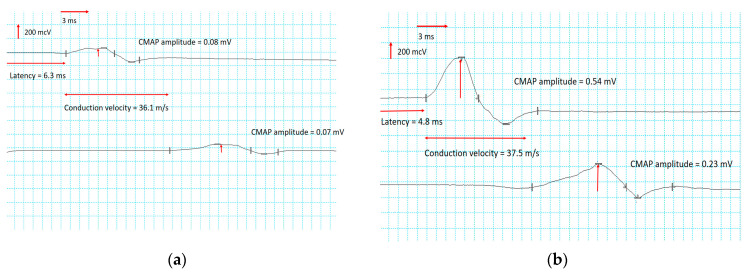
Nerve conduction study of the right peroneal nerve before (**a**) and after (**b**) direct TENS treatment.

**Figure 5 jfmk-09-00040-f005:**
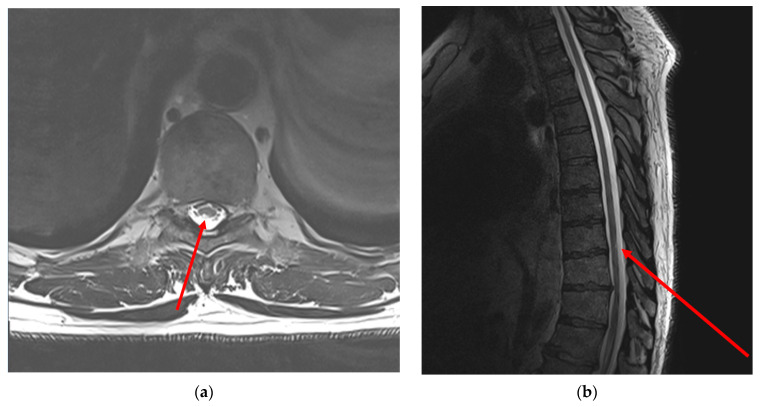
T2-weighted MRI images of the thoracic spine demonstrate incomplete segment of hyperintensity at cervical Th9-Th10 level in axial scan (**a**) and in sagittal scan (**b**). Note: red arrows indicate the affected area of the spinal cord. Neuroimaging examination was performed on the 126th day after the onset of neurological symptoms.

**Figure 6 jfmk-09-00040-f006:**
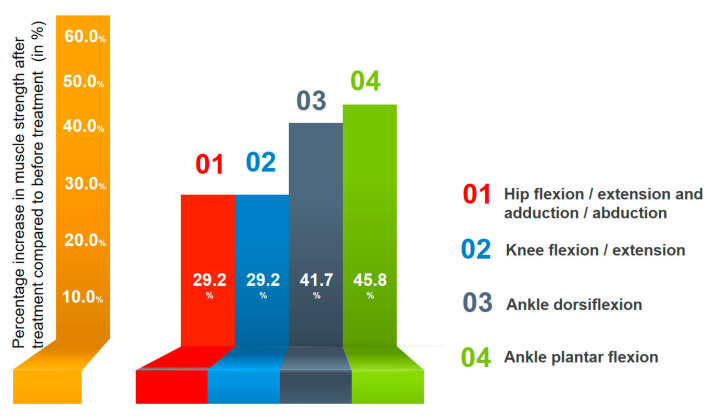
Dynamics of muscle strength of the lower extremities after combined direct TENS therapy compared to initial values before treatment.

**Figure 7 jfmk-09-00040-f007:**
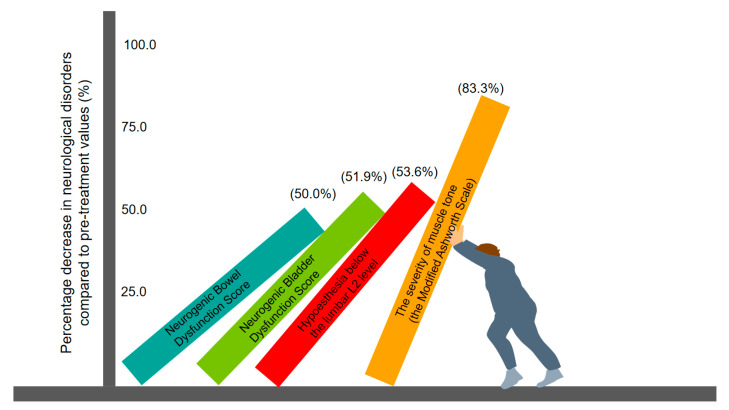
The decrease in neurogenic bowel dysfunction, neurogenic bladder dysfunction, hypoesthesia below the lumbar L2 level and the severity of muscle tone by the Modified Ashworth Scale after direct TENS therapy compared to initial values before treatment.

**Figure 8 jfmk-09-00040-f008:**
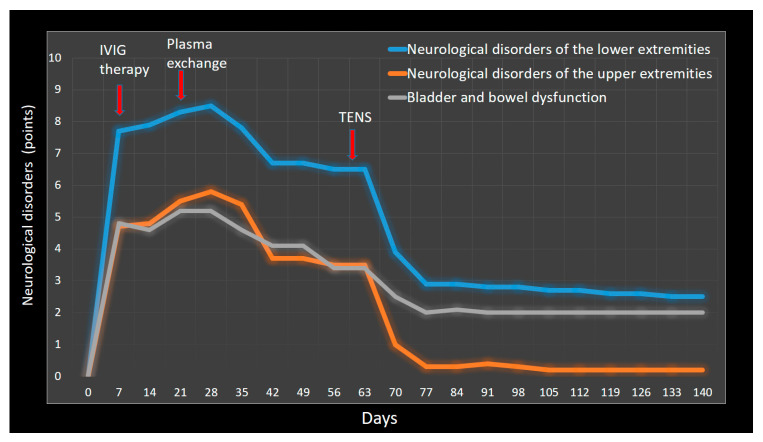
The dynamics of neurological disorders of the lower and upper extremities and neurogenic bladder and bowel dysfunction in observation period in days. Notes: IVIG: intravenous immunoglobulin; TENS: transcutaneous electrical nerve stimulation.

**Table 1 jfmk-09-00040-t001:** Onset and development of the disease before direct TENS treatment.

Date	Anamnestic Data before TENS Treatment
21 May 2022	Contact with people with confirmed coronavirus infection (COVID-19).
28 May 2022	Fever up to 38.5 degrees Celsius with runny nose and cough.
1 June 2022	Positive PCR test for COVID-19.
8 June 2022	No substantive changes in lungs were detected in chest CT scan.
10 June 2022	Burning sensation on the inner surface of the right thigh.
13 June 2022	Spread of burning sensation to right foot.
14 June 2022	Numbness and hypoesthesia in the right leg. Fast progression of weakness in the right leg.
15 June 2022	Numbness and hypoesthesia in the left leg. Fast progression of mild weakness in the left leg.
17 June 2022	Hospitalization in the neurological department with a diagnosis of post-COVID-19 Guillain–Barré and acute transverse myelitis overlap syndrome.
17 June 2022	Neurological disorders: incomplete bladder evacuation, indwelling bladder catheterization, loss of the sensation of rectal fullness and ability to evacuate bowels. Administering of cleansing enema. SF-Qualiveen score 3/4, Neurogenic Bowel Dysfunction Score 10/47. Lower extremities: motor strength 2/5 points, hypoesthesia 9/10 points, neuropathic pain 8/10 points. Upper extremities: motor strength 5/5 points, hypoesthesia 4/5 points, neuropathic pain 6/10 points. Patient could not stand up unaided.
17 June 2022	Treatment was started with intravenous immunoglobulin 0.4 g/kg daily for 5 days. Immunoglobulin therapy was commenced without benefit and progressive clinical deterioration. Then, 10 days after completion of immunoglobulin treatment, plasma exchange 200 mL/kg was started for 5 sessions within 10 days.
10 July 2023	Incomplete course of electromagnetic field therapy (5 sessions). Physiotherapy was excluded due to increased pain, burning and tingling in the legs during procedures.
31 July 2022	Control over voluntary urination was not complete. Intermittent catheterization was performed every 8 h. Bowel care time increased to 1–1:20 h. Mini-enema was used on average once a week. SF-Qualiveen score 2.6/4.0, Neurogenic Bowel Dysfunction Score 8/47. Lower extremities: motor strength 3/5 points, hypoesthesia 9/10 points, neuropathic pain 7/10 points. Upper extremities: motor strength 5/5 points, hypoesthesia 3/5 points, neuropathic pain 5/10 points. Patient could stand up unaided and could walk with aid but not run.
1 August 2022	Hospital discharge.
1 August 2022	Home medication.
17 August 2022	Outpatient transcutaneous electrical neurostimulation therapy.

**Table 2 jfmk-09-00040-t002:** Nerve conduction values of the motor median, ulnar, peroneal and tibial nerves before and after direct TENS therapy.

		Before Treatment			After Treatment			Lower Limit of Normal [36]		
Nerve		CMAP	CV	TL	CMAP	CV	TL	CMAP	CV	TL
Median nerve	Left	10.2	43.0	5.8	7.65	47.2	4.7	>5 mV	>49 m/s	<4.6 ms
	Right	7.76	43.4	5.7	6.34	46.3	4.8	>5 mV	>49 m/s	<4.6 ms
Ulnar nerve	Left	9.90	44.3	5.02	9.10	49.6	4.6	>5 mV	>55 m/s	<4.6 ms
	Right	6.40	40.1	5.20	6.20	45.2	4.9	>5 mV	>55 m/s	<4.6 ms
Common peroneal nerve	Left	0.02	36.8	10.9	0.05	38.0	8.8	>3 mV	>40 m/s	<4.9 ms
	Right	0.08	36.1	6.3	0.54	37.5	4.8	>3 mV	>40 m/s	<4.9 ms
Tibial nerve	Left	0.26	32.3	4.8	1.6	38.1	4.6	>3.5 mV	>40 m/s	<4.9 ms
	Right	0.21	34.3	6.5	1.3	41.6	5.8	>3.5 mV	>40 m/s	<4.9 ms

Note: CMAP: compound muscle action potential; CV: conduction velocity; TL: terminal latency.

**Table 3 jfmk-09-00040-t003:** Nerve conduction values of the sensory median, ulnar, peroneal, tibial and sural nerves before and after direct TENS therapy.

		Before Treatment		After Treatment		Lower Limit of Normal [36]	
Nerve		SNAP	CV	SNAP	CV	SNAP	CV
Median nerve	Left	0.68	37.6	0.77	35.0	>5 mcV	>50 m/s
	Right	0.52	37.2	0.66	38.9	>5 mcV	>50 m/s
Ulnar nerve	Left	0.19	50.4	0.45	52.7	>5 mcV	>50 m/s
	Right	0.97	50.1	1.29	53.1	>5 mcV	>50 m/s
Superficial peroneal nerve	Left	No response		No response		>5 mcV	>50 m/s
	Right	No response		No response		>5 mcV	>50 m/s
Medial plantar nerve	Left	No response		No response		>5 mcV	>50 m/s
	Right	No response		No response		>5 mcV	>50 m/s
Sural nerve	Left	No response		No response		>5 mcV	>50 m/s
	Right	No response		No response		>5 mcV	>50 m/s

Note: SNAP: sensory nerve action potentials; CV: conduction velocity. Sensory studies were performed antidromically.

## Data Availability

Data are contained within the article.

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
