# Peer review of "Functional Recovery and Regenerative Effects of Direct Transcutaneous Electrical Nerve Stimulation in Treatment of Post-COVID-19 Guillain–Barré and Acute Transverse Myelitis Overlap Syndrome: A Clinical Case"

_jfmk, 2024, doi:10.3390/jfmk9010040_

Round 1
Reviewer 1 Report
Comments and Suggestions for Authors
In this manuscript, the Authors describe the application of direct TENS on a patient with overlapping transverse myelitis-Guillain Barré syndrome.
The case is well presented and a lot of details are provided on the clinical presentation, disease course etc. However, there is no indication that TENS has changed the course of the disease, and we cannot exclude that all the improvements (including the delayed improvement in sensory, motor and autonomic functions) may be associated with the immunomodulatory treatment and the physical therapy.
In this patient TENS was a safe procedure but we cannot know if it has been really useful and improved outcome.
Other comments:
- I suggest the Authors to summarize several parts of the manuscript (e.g. the brain MRI is normal, so there is no need to thoroughly describe all the normal results; clinical and neurophysiological results can be better highlighted in tables, expressing all the modifications in the text is not necessary or useful).
- To justify the usefulness of TENS, it could be helpful to compare the clinical course of this case with what is known in literature; some information on transverse myelitis and Guillain Barré syndrome should also be added in the Introduction;
- there should be more details for the timing of CNS imaging, dosage, schedule and timing of i.v Ig and plasma exchange, for how many times TENS was performed, timing of post treatment clinical evaluation.
Author Response
Reviewer 1
First of all, my co-authors and I would like to express our deep gratitude for reviewing our manuscript. No doubt adding more information to our manuscript as per your comments has added a lot of clarity to the readers. Where needed, the added text has been indicated in green and modified text in yellow throughout the manuscript for easy reading.
Comment 1. There is no indication that TENS has changed the course of the disease, and we cannot exclude that all the improvements (including the delayed improvement in sensory, motor and autonomic functions) may be associated with the immunomodulatory treatment and the physical therapy. In this patient TENS was a safe procedure but we cannot know if it has been really useful and improved outcome.
Authors response:
There is no doubt that the combination of plasma exchange with pharmacotherapy was highly effective in treating this patient. We added table 1, which demonstrates changes in neurological status before and after plasma exchange treatment. It is important to note that in the hospital the patient underwent incomplete course of electromagnetic field Therapy (5 session) and was excluded due to increased pain, burning and tingling in the legs during procedures. This information was added in table 1.
To determine the therapeutic effectiveness of TENS against the background of the therapeutic effect achieved after the use of plasma exchange and pharmacotherapy, neurological disorders were studied weekly from the onset of the disease to the end of observation. For ease of comparison, the average values of motor deficit, hyposthesia and neuropathic pain were calculated after conversion to a 10-point system. The mean value of bladder and bowel dysfunction was also assessed after converting the SF Qualiveen scores and neurogenic bowel dysfunction scores into a 10-point system (Figure 8). A decrease in neurological disorders was noted in the lower extremities, upper extremities and bladder and bowel function on the 30th day after the use of plasma exchange with pharmacotherapy - 23.6%, 39.7% and 34.65, respectively and after the use of direct TENS com-pared with the results obtained on the 30th day after plasma exchange and pharmacotherapy 58.5%, 85.7% and 41.2%, respectively. Thus, we can come to the conclusion that TENS enhanced the therapeutic effect of the previously conducted course of plasma ex-change with pharmacotherapy by 89.5%. This paragraph and figure were added to discussion.
Many studies report that the long-term therapeutic effect after plasma exchange can be up to a year. However, the most pronounced therapeutic effect may develop within 30 days after the end of treatment
- [In one trial with 220 severely affected participants, the median time to recover walking with aid after plasma exchange was 30 days [Chevret S, Hughes RA, Annane D. Plasma exchange for Guillain-Barré syndrome. Cochrane Database Syst Rev. 2017 Feb 27;2(2):CD001798. doi: 10.1002/14651858.CD001798.pub3. PMID: 28241090; PMCID: PMC6464100].
- In another trials with 349 patients was shown that plasma exchange significantly decreased motor deficit and recovered the ability to walk with assistance within 4 weeks [Osterman PO, Fagius J, Lundemo G, Pihlstedt P, Pirskanen R, Sidén A, Säfwenberg J. Beneficial effects of plasma exchange in acute inflammatory polyradiculoneuropathy. Lancet. 1984 Dec 8;2(8415):1296-9. doi: 10.1016/s0140-6736(84)90819-5. PMID: 6150321.].
- In McKhann study the median time to recover walking unaided was 32 days for 245 participants with Guillain-Barré syndrome after plasmapheresis. [Plasmapheresis and acute Guillain-Barré syndrome. The Guillain-Barré syndrome Study Group. Neurology. 1985 Aug;35(8):1096-104. PMID: 4022342.].
Comment 2. No need to thoroughly describe all the normal results.
Authors response:
To exclude multiple sclerosis, demyelinating brain lesions, and optic nerve lesions in the differential diagnosis, we included the results of brain MRI and evoked potentials, which were normal. However, we have reduced the description of these results to a minimum.
Comment 3. Clinical and neurophysiological results can be better highlighted in tables.
Authors response:
We completely agree with the reviewer and have added a new table showing treatment results before using TENS and figure demonstrating dynamics of neurological disorders in all treatment period. Neurophysiological data are indicated in tables.
Comment 4. Expressing all the modifications in the text is not necessary or useful.
Authors response:
Indeed, expressing all changes in the text is neither necessary nor useful. So, we removed Table 1 with the detailed modification of TENS. In the manuscript, we specified only the current characteristics and methodology of TENS.
Comment 5.
To justify the usefulness of TENS, it could be helpful to compare the clinical course of this case with what is known in literature; some information on transverse myelitis and Guillain Barré syndrome should also be added in the Introduction.
Authors response:
The Introduction has been completely revised to include new information about acute transverse myelitis and Guillain-Barré syndrome, as well as the use of TENS in the treatment of acute transverse myelitis and Guillain-Barré syndrome.
Comment 6. There should be more details for the timing of CNS imaging.
Author responses:
The first MRI was performed 7 days after the onset of neurological symptoms. Changes on MRI of the thoracic region were unreliable. 35 days after the onset, a re-examination revealed minor changes in the thoracic region, characteristic of acute transverse myelitis. 126 days after onset ( 1 month after TENS therapy) MRI of the central nervous system was repeated. Information about MRI timing has been added to the text in lines 184-185, 307-308.
Comment 7. There should be more detail for dosage, schedule and timing of i.v Ig and plasma exchange.
Treatment was started with intravenous immunoglobulin 0.4 g/kg daily for 5 days. Immunoglobulin therapy was commenced without benefit and progressive clinical deterioration. 10 days after completion of immunoglobulin treatment, plasma exchange 200 ml/kg was started for 5 sessions within 10 days. This information was added to text in table 1.
Comment 8. For how many times TENS was performed?.
Authors response:
Direct TENS was carried out in two parallel courses. In the first course were stimu-lated peroneal and tibial nerves bilaterally, In the second course stimulation of median, ulnar and radial nerves was performed on each side. The number of sessions in each course was 15 times. Each of these courses was conducted alternately every other day. The duration of each procedure did not exceed 40 minutes.
Comment 9. Timing of post treatment clinical evaluation
Clinical evaluation was carried out weekly. The last neurological assessment was carried out one month after the end of TENS therapy. This information was added to text in lines 354-367.
Reviewer 2 Report
Comments and Suggestions for Authors
The authors presented a case of 52 yo patient with GBS and ATM overlap syndrome underwent TENS, reporting clinical outcomes. I would like to congratulate with the author for the illustrated caso. The results are well detailed and the discussion is congruent with findings. Here some suggestions to improve the quality of the paper:
- introduction section should be more concised, explaining the gap of the literature and the goal of the paper.
- conclusions should be one or two sentence defining the main message of the paper.
- please check the consistency of the abbrevations.
- please check writing errors (e.g. syn-drome in the title, dot lack at the end of the conclusion paragraph, and others within the text)
- English revision is needed.
It needs minor revisions.
Author Response
Reviewer 2
We would like to thank the Reviewer for his valuable comments and suggestions which have improved the manuscript. Below, we hope to give satisfactory responses to all the reported concerns.
In the interest of clearness, the Reviewers’ comments are completely replicated below. Where needed, the added text has been indicated in green and modified in yellow throughout the manuscript for easy reading.
Comment 1. Introduction section should be more concised, explaining the gap of the literature and the goal of the paper.
Author response: The Introduction has been completely revised to include new information about acute transverse myelitis and Guillain-Barré syndrome, as well as the use of TENS in the treatment of acute transverse myelitis and Guillain-Barré syndrome. Removed paragraphs between lines 63-78 and lines 87-91.
Comment 2. Conclusions should be one or two sentence defining the main message of the paper.
We have shortened the conclusion in two sentence that define the main message of the paper.
Comment 3. Please check the consistency of the abbreviations.
Thank you for your comment, we re-checked the consistency of the abbreviations and corrected any mistakes.
Comment 4. please check writing errors (e.g. syn-drome in the title, dot lack at the end of the conclusion paragraph, and others within the text).
Thank you again. We checked writing errors in the text and corrected them.
Comment 5. English revision is needed.
English revision has been made.
Reviewer 3 Report
Comments and Suggestions for Authors
Comments and Suggestions for Authors:
The paper that I reviewed is an interesting case study regarding the therapeutic value of TENS in the recovery process from neurological diseases that affect the central and peripheral nervous system. I have some suggestions to make to improve the article:
1) What kind of autoimmune tests did you perform? Have you performed antibodies against autoimmune or onconeural diseases?
2) Taking into account the simultaneous damage to the central and peripheral nervous systems, have you discussed a MOGAD syndrome? Did you perform anti MOG antibodies?
3) On line 197 it is described a decreased onset latency of CMAP instead of increased as it happens in autoimmune diseases like Guillain-Barré syndrome (the action potential has a delay so the onset latency is increased instead of decreased and this is also evident on the ENG that you performed).
4) On line 198 please correct f with of.
5) Did you have a temporal dispersion of CMAP? This is also a criterion for demyelination of the peripheral nervous system and as I can see on the EMG images there seems to be temporal dispersion. Please clarify the aspect.
6) How old is the onset of the disease? You described an acute denervation accompanied by reinnervation aspect in MUAP. Is there an active, ongoing denervation? Is it an acute demyelinating polyradiculoneuropathy or a chronic process?
7) In table no 2 the onset latencies for median, ulnar, left common peroneal nerve are increased more before TENS and still increased but with a certain amelioration after TENS. Please specify this.
8) On line 301 the idea formulated regarding the onset latencies is not clear. Please try to reformulate it.
9) Can you replace the spinal cord MRI images with bigger and clearer one?10) On line 345 please replace Necturnal with nocturnal term.
11) For more discussions regarding the etiology of Guillain-Barré and acute transverse myelitis overlap syndrome please review the following article: Stoian A, Motataianu A, Bajko Z, Balasa A. Guillain-Barré and Acute Transverse Myelitis Overlap Syndrome Following Obstetric Surgery. J Crit Care Med (Targu Mures). 2020, 6(1):74-79. doi: 10.2478/jccm-2020-0008.
Comments on the Quality of English Language
Some minor English editing is required.
Author Response
Reviewer 3
We are very grateful to the reviewer for his very valuable and important comments, especially in the field of electromyography, which played a significant role in improving our manuscript.
Comment 1. What kind of autoimmune tests did you perform? Have you performed antibodies against autoimmune or onconeural diseases?
Authors response:
No abnormalities were detected in routine biochemical and hematological studies, in-cluding serum levels of 25OH(D3), cyanocobalamin, thyroid hormones (T3, T4, TSH), glycated hemoglobin, creatine phosphokinase, and serological tests for hepatitis B and C, human immunodeficiency virus and Syphilis. Detection of neuromyelitis optica IgG was negative with titers less than 1/10 (normal range < 1:10).
Detection of neuromyelitis optica IgG was negative with titers less than 1/10 (normal range < 1:10). Twelve kinds of ganglioside autoantibody were measured, including GM1, GM2, GM3, GM4, GQ1b, GT1b, GT1a, GD1a, GD1b, GD2, GD3, and sulfatide. All these tests were negative. Serum immunoglobulin (IgG and IgM) of AGAs were tested. All these tests were negative. Anti-MOG, anti-aquaporin-4 (anti-AQP4, anti-cardiolipin (CL)/β2, glycoprotein I (β2GPI) testing was negative. Tumor markers were within the normal range. These information’s were added to the text in lines 162-166.
Comment 2. Taking into account the simultaneous damage to the central and peripheral nervous systems, have you discussed a MOGAD syndrome? Did you perform anti MOG antibodies?
Author response:
Thank you for this important question. Yes, we examined Myelin oligodendrocyte glycoprotein antibodies. Result of this examination was negative. This information was added to the text in lines 162-166.
Comment 3. On line 197 it is described a decreased onset latency of CMAP instead of increased as it happens in autoimmune diseases like Guillain-Barré syndrome (the action potential has a delay so the onset latency is increased instead of decreased and this is also evident on the ENG that you performed).
Author response:
Thank you for your very careful study of our manuscript. A whole phrase is missing here. Of course, motor conduction velocity decreased and CMAP terminal latency increased. We have corrected this error in the text.
Comment 4. On line 198 please correct f with of.
Authors response: Error in line 198 was corrected.
Comment 5. Did you have a temporal dispersion of CMAP? This is also a criterion for demyelination of the peripheral nervous system and as I can see on the EMG images there seems to be temporal dispersion. Please clarify the aspect.
Authors response: Thank you again for your comments in electromyoneurography field. Yes as you noted there are an increase of more than 30% of the compound muscle action potential (CMAP) in proximal stimulation of the left tibial and right peroneal nerves. We add more information about temporal dispersion
in lines 192-193: Additional temporal dispersion in the elbow-wrist nerve segment was regis-tered in examination of ulnar nerves.
In lines 201-203: Temporal dispersion was recorded in the ankle-knee segment of the tibial nerves and in the ankle-fibular head segment of the common peroneal nerves
Comment 6. How old is the onset of the disease? You described an acute denervation accompanied by reinnervation aspect in MUAP. Is there an active, ongoing denervation? Is it an acute demyelinating polyradiculoneuropathy or a chronic process?
Authors response:
EMG before TENS treatment was performed 61-63 years after the onset of the disease. Needle electromyography showed signs of denervation and reinnervation activity with evidence of ongoing denervation including fibrillation potentials and positive sharp waves in the tibialis anterior and gastrocnemius muscles bilaterally. Evoked electroneurography and needle electromyography findings are more characteristic of acute inflammatory demyelinating polyneuropathy (AIDP) with secondary axonopathy.
Comment 7. In table no 2 the onset latencies for median, ulnar, left common peroneal nerve are increased more before TENS and still increased but with a certain amelioration after TENS. Please specify this.
Authors response:
Numerous experimental studies have suggested that application of TENS led not only to the acceleration of functional and motor recovery, but also to an enhance in the axon quantity and in the diameter of the regenerated axons. In other studies, have shown that low-frequency TENS, but not high-frequency TENS, results in increased fiber diameter and thickening of the myelin sheath.
Comment 8. On line 301 the idea formulated regarding the onset latencies is not clear. Please try to reformulate it.
Authors response: We have reformatted this paragraph.
Comment 9. Can you replace the spinal cord MRI images with bigger and clearer one?10)
Authors response: Yes, of course we did it.
Comment 10. On line 345 please replace Necturnal with nocturnal term.
Authors response: Thanks, we fixed it.
Comment 11. For more discussions regarding the etiology of Guillain-Barré and acute transverse myelitis overlap syndrome please review the following article: Stoian A, Motataianu A, Bajko Z, Balasa A. Guillain-Barré and Acute Transverse Myelitis Overlap Syndrome Following Obstetric Surgery. J Crit Care Med (Targu Mures). 2020, 6(1):74-79. doi: 10.2478/jccm-2020-0008.
Authors response: Thank you very much for the advice to read this manuscript. In our opinion, the etiology of the GBS and ATM overlap syndrome in our clinical case is a history of COVID-19 14 days earlier. We clarified the etiology of this disease in the title, text and conclusion as post COVID-19 GBS and ATM overlap syndrome.
Round 2
Reviewer 1 Report
Comments and Suggestions for Authors
I have read the manuscript which I've found really improved, especially for the clarity of the "timeline" of various tests and clinical course. I find particularly useful the newly added figure 8, which highlights the radical improvement of patient's symptoms shortly after the beginning of TENS.
My main initial concern (in my opinion in the first version there was no clear indication of the actual beneficial effects of TENS) has been addressed, as were all my other comments Therefore, in my opinion, the manuscript could be accepted for publicationAuthor Response
Dear reviewer 1,
We thank you for your careful revisions and corrections, and for your constructive comments on our manuscript. They certainly allowed us to improve our article.
Kind regards!